# Gastritis, Gastric Polyps and Gastric Cancer

**DOI:** 10.3390/ijms22126548

**Published:** 2021-06-18

**Authors:** Helge Waldum, Reidar Fossmark

**Affiliations:** 1Department of Clinical and Molecular Medicine, Faculty of Medicine and Health Sciences, Norwegian University of Science and Technology, 7030 Trondheim, Norway; Reidar.fossmark@ntnu.no; 2Department of Gastroenterology and Hepatology, St. Olav’s Hospital—Trondheim University Hospital, 7006 Trondheim, Norway

**Keywords:** ECL cell, gastrin, gastric cancer, gastric polyps, gastritis

## Abstract

Gastric cancer is still an important disease causing many deaths worldwide, although there has been a marked reduction in prevalence during the last few decades. The decline in gastric cancer prevalence is due to a reduction in *Helicobacter pylori* infection which has occurred for at least 50 years. The most probable mechanism for the carcinogenic effect of *H. pylori* is hypergastrinemia since *H. pylori* infected individuals do not have increased risk of gastric cancer before the development of oxyntic atrophy. When atrophy has developed, the carcinogenic process continues independent of *H. pylori*. Autoimmune gastritis also induces oxyntic atrophy leading to marked hypergastrinemia and development of ECL cell neoplasia as well as adenocarcinoma. Similarly, long-term treatment with efficient inhibitors of acid secretion like the proton pump inhibitors (PPIs) predisposes to ECL cell neoplasia of a different degree of malignancy. Contrasting the colon where most cancers develop from polyps, most polyps in the stomach have a low malignant potential. Nevertheless, gastric polyps may also give rise to cancer and have some risk factors and mechanisms in common with gastric cancer. In this overview the most common gastric polyps, i.e., hyperplastic polyps, adenomatous polyps and fundic gland polyps will be discussed with respect to etiology and particularly use of PPIs and relation to gastric carcinogenesis.

## 1. Introduction

The prevalence of gastric cancer has had a marked reduction in prevalence during the last decades. However, it is still an important disease, globally being the third cause of cancer death [1]. It is also an interesting disease in that a bacterium, *Helicobacter pylori* (*H. pylori*), is the dominating cause [2]. It is remarkable that we know the main etiology of an important cancer. However, the mechanism by which *H. pylori* causes gastric cancer is still debated, which to some extent hampers a rational prophylaxis. In the present review we will repeat and strengthen the arguments in favour of the role of gastrin in *H. pylori* carcinogenesis [3] and then focus on the different types of gastric polyps as precursors of gastric cancer. We will further discuss the role of proton pump inhibitors (PPIs), another cause of hypergastrinemia, in the pathogenesis of gastric polyps and gastric cancer.

## 2. Gastric Cancer

The stomach is a saccular organ and accordingly luminal growth of tumours does most often not cause symptoms due to obstruction of the lumen at early phases. This is one of the reasons why most gastric cancers are diagnosed at an advanced stage and the prognosis of gastric cancer still poor. Until now, tumours have curiously been divided into cardiac or proximal when occurring in the small cardiac region, and distal when localised in the entire remaining part of the stomach. A classification based upon a such localisation does not take into consideration that the acid producing oxyntic mucosa and the gastrin producing antral mucosa are completely different, which should be included into a classification [4].

There are many histological classifications of gastric cancer, but the classification by Laurén where glandular growth pattern is classified as intestinal type and tumours without glandular growth as diffuse type, seems to represent biological differences as the types do not convert into each other [5] and have had a marked difference in incidence trends over the past decades [6]. The weakness of the Laurén classification is that 15–20 % of the tumours that cannot be classified to either type, and are therefore classified as intermediate. The differences between intestinal and diffuse types found microscopically are also reflected macroscopically, where carcinomas of the diffuse type often induce a more pronounced fibrosis and may be spread submucosally leading to a non-compliant stomach with reduced motor activity, a so-called linitis plastica [7].

Whereas *H. pylori* has been accepted as the major cause of gastric cancer for 30 years [2], the mechanism by which it causes cancer has not been solved despite enormous efforts. It should also be recalled that a bacterial infection besides *H. pylori* has not been claimed to have a direct carcinogenic effect [8]. Moreover, only a small proportion of individuals infected with *H. pylori* develop gastric cancer. The role of bacterial characteristics, including Cytotoxin-associated gene-A (CagA) antigen expression[9], is poorly understood and the search for specific factors of different strains of *H. pylori* predisposing to gastric cancer has not been successful. Interestingly, CagA^+^ *H. pylori* strains seem to induce more inflammation than other *H. pylori* strains [10]. The exact mechanism for the putative role of CagA in gastric carcinogenesis is still disputed [11], but it has been proposed that CagA produces an oncoprotein which could induce malignancy after being incorporated into a cell [12]. Alternatively, exposure of *H. pylori* carcinogen in vitro has been reported to induce DNA damage [13]. However, these two mechanisms presently lack support from convincing in vivo data. Neither has research aiming at identifying specific traits of individuals developing gastric cancer secondary to *H. pylori* infection shown any consistent results. Therefore, neither studies of properties of the infectious agent nor host characteristics have uncovered the mechanism of gastric carcinogenesis due to *H. pylori* infection. The study by Uemura et al. describing that *H. pylori* gastritis did not predispose to gastric cancer before having induced oxyntic atrophy [14] was a breakthrough. Antigenic mimicry between *H. pylori* and the oxyntic mucosa [15] including parietal cells [16] was discussed as a possibility for the development of gastritis in the late nineties, but has not been followed up later. Since *H. pylori* seems mainly to be acquired in childhood [17] and since atrophic gastritis secondary to *H. pylori* requires decades to develop, as indicated by the decline in gastric acid secretory capacity with age [18], the logic strategy to prevent *H. pylori* induced gastric cancer is to test young adults for *H. pylori* by serology and eradicate the bacterium from positive individuals [19].

In addition to *H. pylori* gastritis, the so-called autoimmune gastritis also causes oxyntic atrophy. The cause of autoimmune gastritis is not known, but differs from *H. pylori* gastritis by antral sparing, adult onset, and more rapid progress leading to complete atrophy of oxyntic glands, finally resulting in anacidity and a marked secondary hypergastrinemia. *H. pylori* gastritis, on the other hand, starts in the antrum and spreads orally, often not affecting the entire oxyntic mucosa. Therefore, patients with *H. pylori* gastritis may not be completely anacidic and accordingly have a less pronounced hypergastrinemia, which also could be due to antral gastritis impairing gastrin release from the G cells [20] (Figure 1). Whereas autoimmune gastritis especially leads to ECL cell derived neuroendocrine tumours (NETs) [21] and more seldom to gastric cancer [22,23], the opposite is due to *H. pylori* gastritis where gastric adenocarcinomas is a more frequent result [2] and gastric NETs are infrequent [24,25]. The difference in prevalence of gastric NETs in patients with *H. pylori* gastritis versus autoimmune gastritis is most likely due to the difference in blood gastrin levels. The mechanism of gastric carcinogenesis due to atrophic oxyntic gastritis may either be an expansion of commensal bacteria colonising the stomach due to lack of gastric acidity or hypergastrinemia. Since oxyntic atrophy seems to play a less important role in the pathogenesis of cardia cancers [26,27], it seems unlikely that commensal bacteria have an important role in gastric carcinogenesis in general. However, as stated above autoimmune gastritis causes not only ECL cell NETs, but also neuroendocrine cancers originating from the ECL cell [28] which may have until now often been classified as adenocarcinomas [29]. Gastrin has a trophic effect particularly on the ECL cell, but also a general and less pronounced effect on the other cell types in the oxyntic mucosa [30] presumably by an effect on the stem cell. Interestingly, hypergastrinemia has by far a more pronounced effect on ECL cell proliferation compared with the stem cell as assessed by autoradiography in rats [31].The general effect may be a direct effect by gastrin or secondary to an effect by a mediator secreted from the ECL cell, most likely regenerating (REG) protein [32,33]. If the REG protein is less efficient to induce stimulation of growth of the stem cell compared with the stimulating effect of gastrin on the ECL cell, this could explain that *H. pylori* gastritis is more prone to predispose to adenocarcinomas with a longer latency (Figure 1). Moreover, it seems established that the reduction in gastric acid secretion recorded in older individuals decades ago is not a result of aging per se, but progression of *H. pylori* induced oxyntic atrophy [18].

Intestinal metaplasia has been considered a separate risk factor for developing gastric cancer [34]. However, it is difficult to discriminate the separate role of intestinal metaplasia from oxyntic atrophy in gastric carcinogenesis [35] since intestinal metaplasia always occurs in a stomach with atrophy. Intestinal metaplasia can just be a marker and not a precursor of gastric cancer [36,37]

Inhibitors of gastric acid secretion induce hypergastrinemia depending on their efficacy, reflecting the central role of gastrin in the regulation of gastric acidity [38]. The risk of hypergastrinemia has particularly focused on the most efficient group, the proton pump inhibitors (PPIs). Depending of individual susceptibility, dose and duration of PPI treatment, a variable degree of hypergastrinemia develops leading to ECL cell NETs [39,40,41] via a phase of ECL cell hyperplasia and dysplasia [42]. Long-term profound acid inhibition by PPIs may also induce gastric neuroendocrine carcinomas originating from the ECL cell [43]. Similarly, an inactivating mutation in the gene encoding the alpha sub-unit of the proton pump was reported to induce ECL cell NETs in patients in the third decade and a combined ECL cell NET and adenocarcinoma in the fourth decade [44,45] demonstrating that the ECL cell has the capacity to develop into highly malignant tumours. This has been opposed by some dominating pathologists, for instance by Solcia et al. [46], although their view was different in the late seventies [47]. Moreover, the studies of patients with a genetic predisposition of gastric ECL-cell neoplasia illustrate that large studies following patients for a few years with respect to neoplasia may have observed patients for an impressive number of patient-years [48] but are necessarily inconclusive concerning the long-term risk of cancers. In this context it may seem confusing that PPI treatment for gastroesophageal reflux after *H. pylori* eradication was reported to induce gastric cancer [49,50] and that PPI treatment was reported to increase the prevalence of gastric cancer the first year in a large epidemiological study based upon Swedish registries [51]. The explanation could be that patients having been infected with *H. pylori* have had a long period with hypergastrinemia before starting with PPI which has made them susceptible to a further increase in the gastrin level caused by PPI. In conclusion of this part, *H. pylori*, autoimmune gastritis and PPIs all predispose to gastric cancer via gastrin. The increase in gastric cancer incidence from 1995 described in young Americans after decades of decline [52] may be related to the increasing use of PPIs in the treatment of acid related diseases as well as mild discomfort from the upper abdomen [53].

The interaction between *H. pylori* gastritis and PPI treatment has been recognised for long. Attention was first drawn to this by the report describing progression of atrophic gastritis during PPI treatment [54]. Subsequent studies have given ambiguous results concerning the effect of PPIs on the severity and spread of gastritis [55]. However, there is no doubt that PPI treatment induces a more marked hypergastrinemia in individuals with *H. pylori* gastritis [56], which was to be expected in those with gastritis affecting the oxyntic mucosa. The exaggerated hypergastrinemia caused by PPI use in patients with *H. pylori* infection is a strong indication for *H. pylori* eradication in patients who are anticipated to start long-term PPI treatment. It is remarkable that hypergastrinemia still has not been generally accepted as a central mechanism in gastric carcinogenesis. Thus, every condition with long-standing hypergastrinemia in every species examined predisposes to gastric neoplasia of a different degree of malignancy. This has been demonstrated in rodents including rats, mice and Japanese cotton rats [57,58,59], as well as in man with sporadic gastrinoma [60] or gastrinomas as part of multiple endocrine neoplasia type I [61], autoimmune gastritis [21], *H. pylori* gastritis [25], proton pump inhibitor treatment [40,43], after surgery where antral mucosa is no longer exposed to acidic gastric juice [62] and in families with an inactivation mutation in the proton pump resulting in anacidity [44,45]. The neglection of the importance of gastrin in gastric carcinogenesis may have serious consequences, since the incidence of gastric cancer could be markedly reduced by early eradication of *H. pylori* before development of oxyntic atrophy, which may be achieved by testing and treating young adults. In those with established oxyntic atrophy either due to *H. pylori* or autoimmune gastritis, treatment with the specific gastrin antagonist netazepide [63,64] could be an option. Moreover, long-term treatment of large proportions of the population with efficient inhibitors of gastric acid secretion like the PPIs and the potassium competitive acid blockers (PCABs) should be avoided, particularly in young individuals. There is an increasing number of papers reporting an association between PPI use and gastric cancer [65,66]. This should prompt clinicians to reduce gastric acid secretion just sufficient to allow healing of inflammation and relieve symptoms in long-term treatment of acid related gastro-oesophageal reflux. This is particularly important in young individuals [67]. Treatment of mild to moderate oesophagitis should be started with histamine-2 blockers since their effect is reduced after PPI treatment due to tolerance [68]. There are other less prevalent causes of gastric cancer such as congenital lack of E-cadherin or infection with Epstein Barr virus [10], which will not be covered in this review. There are also other possible mechanisms for development of gastric cancer than just hypergastrinemia. Thus, Epidermal growth factor (EGF) was described to play an important role in regeneration of human gastric mucosoid cultures [69]. It is conceivable that *H. pylori* gastritis through inflammation could affect EGF concentration and thus play a role in carcinogenesis. However, as *H. pylori* gastritis continues to predispose to gastric cancer development many years after eradication or loss of *H. pylori* [70,71] and there is no relationship between *H. pylori* infection and expression of EGF receptor 2 in gastric cancer [72], such a mechanism seems less likely. Similarly, based upon a mouse model of linear tracing, a Lgr5^+^ subpopulation of chief cells activated by Wnt signalling functioned as stem cells after injury, and was claimed to play a role in gastric carcinogenesis localised to the corpus [73]. Since the carcinogenesis continues after loss of *H. pylori* in a stomach with some degree of atrophy [74], this mechanism also seems less plausible. The localisation of gastric carcinomas has been classified into proximal meaning those localised close to the cardia and distal as those localised to the rest of the stomach. Such a classification is very unlogical since the differences between the oxyntic and antral mucosa are ignored. When analysing cancers localised to the antrum and those in the corpus/fundus separately, hypergastrinemia is markedly associated with later development of adenocarcinoma in the corpus/fundus [75]. Furthermore, when using such a classification, it should be remembered that oxyntic glands may be found in the proximal part of the antrum [76]. In either way, in order to improve the understanding of gastric carcinogenesis, the tumour localisation should be carried out according to the three mucosae found in the normal stomach (cardia, oxyntic and antral). The data are overwhelming that oxyntic atrophy and gastrin are central in gastric carcinogenesis. The trophic effect of gastrin on the ECL cell predisposes to mutations by increased number of cell divisions each having an inherited risk of mutation. With time mutations with functional effects may lead to changes in malignant direction [77,78].

## 3. Gastric Polyps

Gastric polyps are lesions protruding into the lumen and are increasingly found at gastroscopy, in Western populations in more than 6% of patients [79,80]. Polyps may develop in all parts of the stomach, have a heterogeneous origin from different cells and tissues, and the different subtypes have a highly variable prevalence. In this review we will concentrate on polyps of epithelial origin being the most prevalent and discuss their malignant potential. Gastric polyps may be sporadic or develop in individuals with genetic syndromes predisposing to polyps and cancers in several organs. Based upon their histomorphology gastric polyps may be divided into hyperplastic polyps, adenomatous polyps and fundic gland polyps. We will focus on their etiology and particularly their relationship to *H. pylori* as well as PPI use and their risk for malignant transformation. We will also briefly mention polyps occurring in some patients with liver cirrhosis and portal hypertension.

### 3.1. Hyperplastic Polyps

Hyperplastic polyps (HPs) are prevalent and develop in association with gastritis and gastric atrophy [81,82] or may develop in response to injury that stimulates regeneration and proliferation. The histological appearance of hyperplastic polyps overlaps with polyps that have previously been characterised as inflammatory, the latter term now considered a misnomer [83]. HPs are most often single and may be found in all parts of the stomach. Previously, HPs were most often found as single polyps in the antrum, whereas there has been described a shift towards a more proximal location for HPs as well as for other gastric polyps [84]. Sporadic hyperplastic polyps cannot be distinguished morphologically from polyps occurring in syndromes with hyperplastic polyposis [79]. The sporadic HPs are associated with gastritis and atrophy, the same conditions being central in gastric carcinogenesis discussed above. Hyperplastic polyps have repeatedly been found to regress after *H. pylori* eradication, demonstrating the causal role of *H. pylori* [85]. There is an increased risk of gastric neoplasia in patients with HPs [86]. The mechanism by which HPs develop through presumed proliferation of the foveolar surface mucosa is not known, but since they are localised, there must be a local change in the growth regulation or a mutation in the cell of origin affecting the proliferation rate. HPs also develop in the cardia [87] of patients with gastroesophageal reflux disease (GERD). Hypertrophic gastropathy in Ménétriers disease, a condition of uncertain etiology, but with increased risk of gastric cancer development, may also contain elements that mimic HPs [88].

Since oxyntic atrophy is an important predisposing factor, gastrin could also play a role. The occurrence of HPs not only in the oxyntic mucosa, but also in the antral and cardiac mucosae illustrates that gastrin does not seem to be the only etiological factor [81]. However, during the recent years it has become clear that oxyntic glands can be found deep in the antral mucosa [76]. In patients HPs disappeared after *H. pylori* eradication gastrin was reduced compared with patients with persisting HPs, suggesting that gastrin was the pathogenic factor [89]. Moreover, there is an interesting description of neuroendocrine cell hyperplasia and gastric NET within gastric HPs [90]. Hypergastrinemia is associated with hyperplastic polyps [91], which has been observed in both cross-sectional studies as well as in prospective follow-up. Interestingly, fundic argyrophil hyperplasia which reflects hyperplasia of ECL cells and thus hypergastrinemia, was already in 1984 described to be associated to HPs [92]. A trend towards hypergastrinemia as risk factor for malignant progression of hyperplastic polyps has also been reported [93]. The prevalence of dysplasia in sporadic hyperplastic polyps has been reported to be in the range 1.9–19% [94], with a marked association with size [95]. Adenocarcinomas may develop from hyperplastic polyps and are found in up to 2.1% of resected polyps [96]. The risk is associated with size, an observation that also has reached clinical guidelines with recommendations of removal of polyps lager than 5 mm [97] or 10 mm [94]. A similar finding of NET in a large HP with adenocarcinoma development in a patient with hypergastrinemia was recently reported [98]. The positive trophic effect of mediators released from the ECL cell, like REG protein [20,21] could induce the growth resulting in polyp formation and thereby linking the occurrence of HPs to oxyntic atrophy and hypergastrinemia. There are also two reports of signet ring cell carcinoma in hyperplastic polyps [99,100]. Interestingly, there is also a case report describing development of adenocarcinoma in multiple hyperplastic polyps in a patient with long-term use of PPI and *H. pylori* infection leading to marked hypergastrinemia [101]. In this report there was also ECL cell micronests at the base of the polyps. Hyperplastic polyps have also been described in a patient on PPI without *H. pylori* infection [102]. HPs may also develop in patients with portal hypertension [103]. In a patient with portal hypertension due to liver cirrhosis, long-term PPI treatment resulted in HPs which disappeared after discontinuation of PPI [104]. PPI use may also cause hyperplastic polyposis in the oxyntic mucosa of *H. pylori* negative patients [102] which regress after cessation of PPI use. In conclusion, hyperplastic polyps have been found to be markers of an increased risk of carcinoma development elsewhere in the stomach, which may be explained by their close relationship with *H. pylori* infection, atrophy and hypergastrinemia. Accumulation of ECL cells may be one of the causes of hyperplastic polyps and thus gastric hypoacidity via hypergastrinemia could play a role in the pathogenesis.

### 3.2. Adenomatous Polyps

Adenomas are precursors of gastric adenocarcinomas and frequently arise on a background of atrophic gastritis. The risk of malignancy is associated with size [105] as well as histological subtype. The gastric adenomas are subclassified based upon morphology [79], where adenomas of intestinal type and fundic gland type have a higher risk of progression to carcinoma than foveolar and oxyntic gland adenomas [54]. Foveolar type adenomas are most often solitary, small and they seldom develop into malignancy. The intestinal type adenomas, which are recognised by specific intestinal cells like goblet and Paneth cells, generally develop in a mucosa with atrophic gastritis whether due to *H. pylori* or autoimmune gastritis, and they have a malignant potential. *H. pylori* may play a role in progression of adenomas to adenocarcinomas of intestinal type as eradication of *H. pylori* is associated with lower risk of gastric cancer during follow-up [105]. The mutations found in gastric adenomas vary, but many are also found in gastric cancers, including *kras* and *beta-catenin* [106]. As for hyperplastic polyps there is an association between gastric adenomas and synchronous as well as metachronous adenocarcinomas. This has been reflected in clinical practice guidelines with recommendations of *H. pylori* eradication, thorough inspection of the non-polypoid mucosa and follow-up endoscopies after resection of the adenoma [72]. Patients with gastric adenomas have higher gastrin levels [107], which may be explained by the close relationship with atrophic gastritis in the surrounding mucosa. It has not been shown that PPI treatment predisposes to development of gastric adenomatous polyps.

### 3.3. Fundic Gland Polyps

Fundic gland polyps (FGPs) are the most prevalent type of gastric polyps in recent studies of Western populations [79] and have been found in up to 5% of patients undergoing upper endoscopy [80]. FGPs may occur in patients with familial adenomatous polyposis (FAP) or be so-called sporadic. Histologically there is no difference between FGPs being part of FAP or the sporadic ones [108]. Although FGPs generally are regarded as benign lesions, proliferative cells are observed at the base not only in the proliferative area, but also in epithelium lining the cysts [108]. Moreover, gastric cancer may develop from FGP in patients with FAP [109], and high-grade dysplasia has been reported even in sporadic FGPs [110]. Most sporadic FGPs have *beta-catenin* mutations [111] but very rarely contain dysplasia, whereas FGPs in patients with FAP contain a *APC*-mutation and dysplastic FGPs often contain a second hit *APC* mutation, but no *beta-catenin* mutation, which is believed to cause dysplasia and a risk of adenocarcinoma development. These frequently found mutations in the Wnt-pathway indicate that all FGPs should be considered neoplastic polyps [79]. In a large study population the patient age was lower and the incidence higher for development of dysplasia in FGPs in patients with FAP than in patients with sporadic FGPs [112]. Patients with FAP seem to have an increased risk of gastric adenocarcinoma [113,114], which may previously have been underestimated in Western populations [115], whereas the risk may be 2–4 times higher [113] in Asian patients. Patients with gastric adenocarcinoma and proximal polyposis of the stomach (GAPPS) have fundic gland polyposis [116,117]. This syndrome is considered a variant of FAP as patients have a mutation in the non-coding part of the *APC* gene that predisposes to this particular phenotype. The proximal stomach may be carpeted with polyps and distinct protruding FGPs are common [118]. Of concern is a description of a recent increase in the prevalence of gastric cancer in patients with FGPs connected to FAP [114]. In a large study Genta et al. found no association between occurrence of sporadic FGPs and gastrointestinal malignancies [119]. However, such an association to gastric cancer might have been missed since the prevalence of *H. pylori* was significantly reduced in those with FGPs. In fact, Helicobacter pylori infection has a strong negative association with occurrence of sporadic FGPs, and FGPs hardly develop in *H. pylori* infected individuals [120]. In general, dysplasia in FGPs may not be so seldom as previously thought [121]. Interestingly, PPI treatment induces FGPs [122,123] in a time and dose-dependent manner [122,124], whereas *H. pylori* infection protects against development of FGPs [122,125]. Although PPI treatment may cause hypergastrinemia, the induction of FGPs by PPI is not associated with the degree of hypergastrinemia [122]. The mechanism by which PPIs induce FGPs is not known, but may be related to stagnation of fluid in the oxyntic glands causing cystic dilations [126]. Anyhow, PPI induced FGPs disappear when stopping treatment [127,128], for instance after anti-reflux surgery [129]. FGPs due to PPI treatment were reported to have higher percentage of proliferating cells assessed by Ki67 than other sporadic FGPs [130]. Similarly, three FGPs that were developed after 8 years of PPI therapy showed low grade dysplasia and in one adenocarcinoma [131]. We have also described a tumour growing as polyp localised to a hiatus hernia that was diagnosed after about 15 years of PPI treatment due to gastrointestinal reflux and histologically was classified as a neuroendocrine carcinoma [43]. Anyhow, the risk of gastric cancer in sporadic fundic gland polyps is low and the value of endoscopic polypectomy and follow-up is uncertain. However, in patients with large or multiple polyps, the indication for PPI use should be re-evaluated and PPI therapy stopped or reduced. Endoscopic resection of the largest polyps to confirm the diagnosis and a follow-up may be justified.

### 3.4. Gastric Polyps in Patients with Portal Hypertension

The presence of gastric polyps in patients with liver cirrhosis with portal hypertension has been known for some time, and in 2016 they were described as a new entity [132]. These polyps are of hyperplastic type, but they may be suspected from macroscopical appearance with reddish sausage like folds with prominent capillaries on the surface [104]. Such polyps are actually not so rare as in a large series they were found in one third of patients with cirrhosis [133]. Polyps related to portal hypertension may be so large that they may cause obstruction [134]. Moreover, polyps related to portal hypertension may be induced by PPI treatment as the polyps in such a patient disappeared after stopping treatment [104]. It is probable that patients with liver cirrhosis are more susceptible to PPI side effects including hypergastrinemia due to reduced drug metabolism [135].

## 4. Conclusions

In contrast to the colon where cancers most often develop from polyps, gastric polyps are relatively less important as precursor lesions of gastric cancer. However, gastritis leading to oxyntic atrophy is central in gastric carcinogenesis and hyperplastic polyps and adenomas often develop in atrophic mucosa. Since hypergastrinemia, whether due to oxyntic atrophy or hypoacidity caused by proton pump missense mutation or prolonged use of PPI, predisposes to gastric malignancy, hypergastrinemia seems to be the carcinogenic factor in common. Hypergastrinemia is associated with hyperplastic polyps as well as adenomas, whereas its role in fundic gland polyps is uncertain. The inflammation per se causing the oxyntic atrophy in the initial phase in *H. pylori* gastritis, seems to have no further role in the carcinogenesis. In any way, to eradicate the major cause of gastric cancer, *H. pylori*, at young age and reduce long-term use of efficient inhibitors of gastric acid secretion, will probably lead to a marked reduction in the prevalence of gastric cancer.

## Figures and Tables

**Figure 1 ijms-22-06548-f001:**
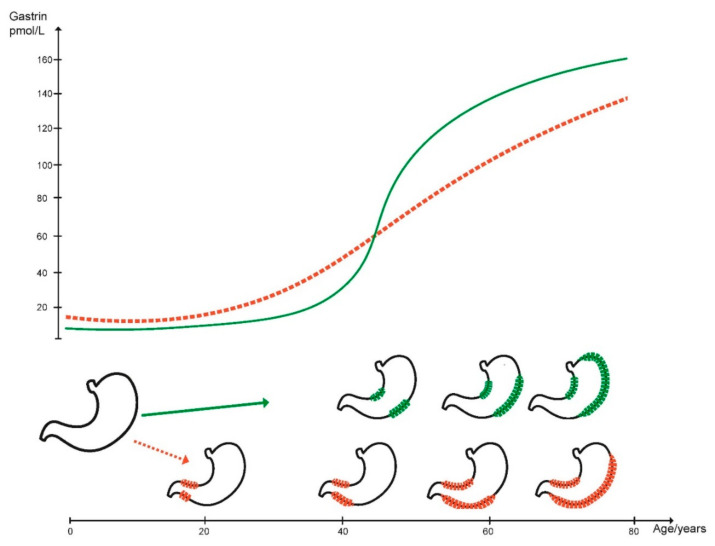
Patients with *H. pylori* (red line) gradually develop hypergastrinemia after infection in childhood, whereas patients with autoimmune chronic atrophic gastritis develop oxyntic atrophy and secondary hypergastrinemia more rapidly in adulthood (green line).

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
