# Peer review of "Gastritis, Gastric Polyps and Gastric Cancer"

_ijms, 2021, doi:10.3390/ijms22126548_

Round 1

Reviewer 1 Report

The present review analyzes the topic of gastric cancer and gastric polyps. Some points need to be clarified in order to make the paper more appealing and useful also for clinician purposes. Remarks are:

  1. Intestinal metaplasia is not considered as a precancerous lesion, it would be worthwhile to include in the manuscript some sentences regarding this condition
  2. Are fundic glands polyps suitable of develop cancer? What must be the clinician approach in the presence of these conditions? Is there a follow-up scheme?
  3. What about the role of antigenic mimicry between Helicobacter pylori and parietal cells in the development of oxyntic atrophy?
  4. “Similarly, long-term treatment with efficient inhibitors of acid 22 secretion like the proton pump inhibitors (PPIs) predisposes to ECL cell neoplasia”. This sentence needs to be supported by clinical clear suggestions about PPI therapy timing and dosage, because it represent a relevant alarm for clinical practice.

Author Response

Thank you for important suggestions. Our answers are:

  1. We have added a paragraph about intestinal metaplasia: Lines 143-147.
  2. Fundic gland polyps may develop into malignancy, but the risk is low. Follow-up scheme: Lines 391-396
  3. Mimicry is discussed: Lines 102-104
  4. PPI and cancer. Clinical suggestions: Lines 210-216 and 391-396.

Reviewer 2 Report

The manuscript describes the role of polyps in gastric cancer. The part about polyps is interesting and well written. However the first part of the paper is about atrophic gastritis  and the role of gastrin. This is also by itself an interesting topic, but the two parts are not connected.

There are some problems:

1) Atrophic gastritis is a lesion, being part of the correa cascade, it would be nice to describe this in the right context

2) The authors describe the link between gastrin ans atrophic gastritis as THE causative event for gastric cancer - certainly this is one way to see the process of carcinogenesis, but likely not the only one - there is much more to discuss - what about the role of EGF/BMP signaling for example (Bocellatto group Gastroenterology 2021), Wnt signaling, what about carcinogenic mutations driving cancer (without atrophic gastritis; (see extensive data by Nick Barker for example). Finally most cancers arise in the antrum, while AG is seen in the corpus as a risk lesion, so increased gastrin concentration could be a co-incidence. How does gastrin cause mutations? 

Together this part would benefit from a revision, to provide several additional aspects on gastric carcinogenesis.

3) The most important point is the missing link between atrophic gastritis and polyps  - are there two distict ways that lead to gc - or are they linked - if yes, how? Or are these two different pathways, if so- this should be reflected in the title and abstract (e.g. atrophy and hyperplasia - two features that promote gc..)

4) H. pylori - there is a lot of data on direct interaction with stem cells, stem cell responses to infection etc, that could be discussed. Direct role of H, pylori and CagA as mutagen should be discussed (recent papers by Hatakayama and Anne Müller for example). H. pylori has many more well described properties that triggering hypergastrinemia. 

Figure: This is a nice figure displaying kinetics of gastritis, but again this does not reflect the content that the title proposes.

Minor: several typos, eg.

Pasients with H. pylori 

Author Response

Thank you for your  suggestions and remarks. Particularly  that the title should be changed and that the two parts (gastritis and gastric polyps) are not connected. We have changed the title accordingly and have made changes in the abstract (line 24) as well as conclusion (lines 413-414, 417-418) to focus more on polyps and as well as common  factors in the pathogenesis (line338-339). We hope that the connection between the two parts is improved. It must also be added that  gastritis is quantitatively more important than gastric polyps in the pathogenesis of gastric cancer.

1.Intestinal metaplasia as a part of Correa cascade is discussed: Lines 143-147

2. Other aspects than gastrin in gastric carcinogenesis is discussed: Lines 218-232. New references 69, 72, 73 are included.. It must be a misinterpretation that most cancers start in the antrum. Could it be that the presently used classification into proximal and distal location with the distal location including both corpus/fundus and antrum could have caused a misunderstanding?  Lines  232-242. Gastrin stimulates the proliferation of the ECL cell, and since each cell division induces a small risk of mutation, every mutagen will increase the risk of tumour: Lines 243-246..

3.  In the original manuscript an association between hyperplastic polyps  with gastritis and gastric atrophy was described with two references. In fact  the role of gastritis and hypergastrinemia in the pathogenesis of hyperplastic polyps was discussed  in the whole section on hyperplastic polyps . Similarly, in the first sentence in the paragraph on adenomatous polyps atrophic gastritis was mentioned. For fundic gland polyps the associations are less clear, but are thoroughly discussed. Hyperplastic polyps are also mentioned in the conclusion to be associated with hypergastrinemia.

4. Other mechanisms than gastrin mediating the carcinogenic effect of H. pylori gastritis are discussed: Lines 84-90, 218-232.

We appreciate that the referee liked the figure. By the change in title and content we hope that the figure now is more suitable.

Round 2

Reviewer 1 Report

None